# Healthcare Professionals’ Attitudes towards and Knowledge and Understanding of Paediatric Palliative Medicine (PPM) and Its Meaning within the Paediatric Intensive Care Unit (PICU): A Summative Content Analysis in a Tertiary Children’s Hospital in Scotland—“An In Vitro Study”

**DOI:** 10.3390/healthcare11172438

**Published:** 2023-08-31

**Authors:** Satyajit Ray, Emma Victoria McLorie, Jonathan Downie

**Affiliations:** 1Royal Hospital for Children, Glasgow G51 4TF, UK; jonathan.downie@ggc.scot.nhs.uk; 2The Paediatric Palliative Care Research Group, Health Sciences, University of York, York YO10 5DD, UK; 3Children’s Hospices Across Scotland (CHAS), Edinburgh EH14 1LT, UK

**Keywords:** attitudes, knowledge, meaning, understanding, paediatric, palliative, intensive, collaborative, tertiary, qualitative research

## Abstract

**Background:** Paediatric palliative medicine (PPM) is a holistic approach to care for children and their families. Services are growing and developing worldwide but significant disparity in service provision remains. The Paediatric Supportive and Palliative Care Team (PSPCT) at the Royal Hospital for Children in Glasgow was established in 2019, but there is still no clear integrated role within the paediatric intensive care unit (PICU) at present. Through analysing the attitudes, meaning, knowledge and understanding of PPM in the PICU environment, we hoped to explore the experiences of those providing paediatric palliative care and to identify any barriers to or facilitators of integrated working to gain a better understanding of providing this care. **Methods:** This qualitative study used a survey composed of five open-ended and five closed questions. Sixteen out of a possible thirty-two responses (50%) were accrued from PICU healthcare professionals, including consultants (n = 19), advanced nurse practitioners (n = 4) and band-seven nurses (n = 9). The data were comprehensively studied and analysed by two coders using summative content analysis with assistance from data management software. Codes were further developed to form categories and subcategories. **Results:** Two categories were found: (1) the role of palliative care and (2) experiences of providing palliative care. A total of five subcategories were found, demonstrating that the PSPCT can enhance care in PICU through collaborative working. Barriers identified included staffing, funding and stigma around palliative care. **Conclusions:** This study shows that PICU professionals have a good understanding of the concepts of PPM and view it as an essential part of PICU work. Barriers related to resources and misperceptions of palliative care can be overcome through improved education, funding and staff retention, but this would require buy-in from policymakers. The perspective from our relatively small team increases generalizability to growing teams across the country.

## 1. Introduction

Paediatric palliative medicine (PPM) is a holistic approach to delivering care to children with serious illnesses and their families [1]. The delivery of palliative care is a global health issue as, despite policy interventions, development remains limited by a lack of investment, recognition and research into supporting its global growth [2]. A UK-wide study demonstrated that, due to advances in medicine, more seriously ill children who would have previously died in infancy are now living beyond their expected prognoses [3], meaning there is a greater need for PPM.

To assist in the management of children with serious illnesses and their families, the integration of PPM across healthcare settings has been increasing. However, there are still disparities amongst services within the UK and worldwide [4]. Despite growing numbers of PPM services being established within paediatric centres in high-income countries [5], how they are integrated is not standardised or agreed upon. This results in implications such as inconsistencies in timely referral and challenges in establishing the ideal PPM service model within a cure-driven paediatric intensive care unit (PICU) [6]. It is thought that paediatric palliative care can improve PICU patients’ quality and length of life [7,8,9], particularly those who are critically ill. This is even more pertinent in modern medicine when many PICU patients now have dependence on technologies and are subject to lengthy recurrent PICU admissions and interventions that carry a high risk of morbidity [10,11]. Many of these patients are spending lengthy periods of time in hospital, often in PICUs, which would suggest that many patients, particularly those with medical complexities and life-limiting and/or life-threatening conditions, would benefit from PPM involvement throughout their disease trajectory, as this has shown to reduce mortality, transfer patients from a PICU setting, and provide cost savings to hospitals [12].

A previous study from one of the largest PICUs in the UK [13] explored the barriers to, knowledge about and attitudes toward referral to paediatric palliative care services. It concluded that parent-related factors and possible physician association of palliative care as relating to death were potential barriers impacting referral to PPM services. Although the PPM evidence base in the UK is growing, it remains limited and requires further exploration. The current literature describes various service models, mainly from the USA, such as the consultative (specialist in PPM providing external expert consultation), integrative (palliative care principles are practiced routinely in the PICU as the standard of care), mixed and embedded (where a specialist in PPM works within the PICU to promote access and provision) models [14]. Originally reviewed in 2010 [9], further guidance has since been released by the World Health Organisation in 2018 [15]. However, there is still not enough evidence to determine an agreed worldwide model [16], as disparities in service provision and resources are evident. More recently, in the USA, a champions-based model has been proposed within cardiac intensive care to allow staff to gain additional expertise in PPM through training in hopes of improving integration [14]. In a UK context, the literature on ideal palliative care service models is scarce as these vary depending on available resources, often being dependent on charity funding and the enthusiasm of individuals [17].

Due to the limited UK-based literature and recent developments in PPM, this study hoped to capture attitudes towards palliative care integration within a PICU context to help inform future service models. “Recent developments” refer to a Scottish tertiary hospital as, in 2019, the Paediatric Supportive and Palliative Care Team (PSPCT) was established and has been working in collaboration with PICU staff on a voluntary basis. Therefore, the aims of this study were to collect data around experiences and the role of PPM within a PICU context and to identify any barriers and facilitators to help inform service development from a UK perspective, adding to the limited literature.

## 2. Materials and Methods

### 2.1. Design and Participants

This article draws on the qualitative findings from a mixed-methods study using a survey via SurveyMonkey containing a total of five free-text questions. The aim of the phenomenological qualitative study was to explore the experiences of those providing paediatric palliative care and to identify any barriers or facilitators.

Purposeful sampling was used to recruit healthcare professionals working in a Scottish tertiary children’s hospital involving a recently established palliative care team, the PSPCT. Healthcare professionals included consultants, advanced nurse practitioners and senior nursing staff working in the PICU. No healthcare professionals surveyed were specialists in paediatric palliative medicine or part of the PSPCT. Healthcare professionals were recruited through e-mails detailing the survey and study information between September 2021 and October 2021. Respondents were sent weekly reminders via e-mail or verbally prompted by SR.

Due to no patient data being collected, ethical approval was not necessary. However, written consent was obtained for involvement and potential future publication. All identifiable information was later anonymised. Prior to completing the survey, participants received an introductory e-mail outlining the way participant data would be used. Following the Research Ethics Committee within the UK Health Departments’ Research Ethics service, an ethical review is not normally required for research involving healthcare or social care staff by virtue of their professional role.

### 2.2. Data Collection

The survey was designed by two members of the research team (JD and SR), both with clinical experience and influenced by the recently developed palliative care team known as the PSPCT. The free-text questions can be found in Table 1.

### 2.3. Data Analysis

After informed consent was received, the survey was later anonymised, and responses were uploaded onto NVivo^12^. The data were analysed using summative content analysis [18,19] to identify frequencies and descriptions of providing palliative care. The analysis process involved a total of five steps: (1) responses were read and re-read multiple times by EVM and SR to help gain familiarity with the data, noting initial thoughts and potential bias; (2) after becoming familiar with the data, SR applied a series of latent codes with regular input from EVM; (3) SR began to note any initial categories, frequently meeting with EVM to discuss any disagreements or thoughts; (4) after a series of revisions, SR formed categories and later presented initial thoughts to EVM; (5) finalised categories were formed, named and discussed with the team (EVM and SR). Throughout the analysis process, frequent team discussions were held to reduce potential bias.

## 3. Results

Originally, 32 healthcare professionals were contacted to take part in the study. However, a total of 16 responses were received. The sample details are shown in Table 2. Due to an oversight when using the survey software, advanced nurse practitioners and consultants were grouped together in their responses.

### 3.1. Categories

A total of two categories found were found, each containing subcategories. The first category, titled “role of palliative care”, contained two subcategories: (1) “collaborative working” and (2) “family-centred care”. The second category, “experiences of providing palliative care”, contained three subcategories: (1) “education and collaboration”, (2) “stigma associated with palliative care” and (3) “resources impacting palliative care”. Originally, we had asked a total of five open questions. However, the findings that came from the data appeared to only relate to four of those questions. The categories and subcategories and their supporting codes and the questions they related to, along with frequencies, can be found in Table 3.

The section below further explores the categories and their supporting quotations found.

#### 3.1.1. Category One: Role of Palliative Care

In their responses, healthcare professionals reflected upon their understanding of palliative care and the role that it plays. The majority focused on aspects such as collaborative working and its holistic nature. Each of the roles was divided into two subcategories, as further explored below.

Subcategory One: Collaborative working

In their responses, several healthcare professionals emphasised the importance of collaborative working, viewing it as a dominant role (see Table 3) when providing palliative care. Respondents included descriptions of this, commenting that it was “integrated” (N1) and a “collaborative approach” (D/ANP2) involving “parallel planning” (D/ANP3, D/ANP7) when possible. Each of the components involving collaborative working was viewed as a vital role in palliative care, with some specifically referencing other team members. Many of the healthcare professionals referenced the newly developed palliative care team and intensivists as key collaborators when providing this type of care.

Subcategory Two: Family-centred care

Healthcare professionals expressed the importance of providing family-centred care, meeting the wishes of the patient and their family members when possible, viewing it as another vital role when providing palliative care. On a more individual (patient) level, healthcare professionals viewed this as helping to meet their patients’ wishes, focusing on “quality of life” (N3, D/ANP3) as it was important that they could make the “most of the time they have left” (D/ANP8). To do so, professionals were expected to “listen” (N5) to patients and ensure they are “compassionate” (D/ANP10), regardless of the setting, such as a hospital, hospice or at home. On a more family level, many professionals mentioned the involvement of family members, one individual describing the palliative care process as helping the “child and family to decide how to best get the most of the time they might have” (D/ANP8), emphasising the importance of listening to each individual and their families. Ultimately, providing family-centred care was thought to consist of meeting both medical and non-medical needs. Medical needs referred to treatment and symptom control, whereas non-medical needs referred to the needs of the patients and their families.

#### 3.1.2. Category Two: Experiences of Providing Palliative Care

Throughout their responses, many participants reflected upon their experiences of providing palliative care and the ideal service model, emphasising the importance of education, identifying stigmas surrounding palliative care and a series of limited resources. All of these were thought to have an impact when providing palliative care.

Subcategory One: Education and collaboration

When reflecting upon their previous experiences of providing palliative care in the PICU setting, respondents did not always have some sort of formalised education in this topic area. This was shown when reflecting upon barriers when delivering palliative care. One respondent described such barriers as resulting from a “lack of knowledge” (D/ANP1) and another referred to “education surrounding the role of palliative care and mindset” (N3). Therefore, it was thought that delivering “education to the team” (D/ANP1) would be highly beneficial.

The role of formalised education was not the only avenue that healthcare professionals reflected upon or viewed as a potential facilitator of providing this type of care, and many made suggestions of collaborating with others. Previously, in category one, the concept of collaboration was viewed as a key component of providing palliative care. Despite this, areas of improvement were seen to be necessary, as some professionals were restricted according to their speciality and pathway, as illustrated in the extract below.


*“How can you ask a cardiologist to give a balanced approach to a single ventricle pathway when they will only manage one side of that pathway?”*
(D/ANP6)

Therefore, to provide effective palliative care and to improve current working practices, healthcare professionals thought that collaboration was an area in which team members could learn from one another. There were suggestions such as involving the palliative care team in “ward rounds” (N2), while another was “regular meets to discuss potential referrals” (D/ANP3) to assist in a more “joined up” way of thinking (D/ANP6).

When asked about service delivery models, the responses were mixed, with some advocating for PICU staff to have a “special interest” (D/ANP1) or be “dual-trained” (D/ANP9), whilst others were clear that the palliative care team were considered “a separate specialty” (D/ANP8) and that they welcomed their input into the PICU as an “accessible, collaborative” (D/ANP10) team.

Subcategory Two: Stigma associated with palliative care

Healthcare professionals believed that there was a stigma associated with providing palliative care, ranging from societal, family and staff fears to a general misconception of what palliative care is. It was thought that, as a society, death is not often talked about, which may lead to a “lack of understanding from patients and parents” (D/ANP10). Family members were thought to sometimes not want to “involve” the “palliative care” team (D/ANP11), leading some staff to fear potentially “upsetting relatives” (N2).

On a staff level, some were fearful when providing palliative care, as some “specialities” were described as “scared” (D/ANP6), whilst others were seen as “more open”, such as “intensivists” (D/ANP3), in comparison to other specialities. It was understood that professionals may shy away from “hard conversations” (N3) and, for some, it was failure in recognising the “inevitable” (D/ANP5). Avoiding such conversations was heavily influenced by the stigmas identified, as some families did not want to involve palliative care, whereas some professionals wanted to avoid the risk of upsetting families.

Subcategory Three: Resources impacting palliative care

There were more practical implications impacting palliative care, as resources such as funding and staff availability were mentioned throughout. The ward was thought to be a fast-paced unit with “high acuity” and lots of “competing patient demands” (D/ANP4), meaning that referral to the palliative care team and/or providing palliative care may not always take priority. There were other implications, such as a lack of staff; for example, the palliative care service was described as unrealistic and unfair, as illustrated in the extract below.


*“Relying on one palliative care consultant which is not a fair way to run a service”*
(D/ANP9)

Staff were thought to be under pressure, lacking in the “manpower” (D/ANP6) and time needed to provide effective and meaningful palliative care. This was worsened by the limited funding available, with one professional viewing this as an obvious barrier, describing it as “the usual—staff, funding” (N5).

## 4. Discussion

It is hoped that the results will assist in future implementation of services and provide us with a better understanding of providing this care.

This qualitative study provides further insight into the perceived role of palliative care from the perspective of PICU professionals. The findings demonstrate that PICU professionals have a good understanding of the principles of palliative care and recognise it as a core element of PICU working, differing from previous studies that have demonstrated misperceptions around palliative care, potentially impacting and limiting care that patients and families should receive [20,21,22]. None of the respondents in our study thought that palliative care was only about end-of-life care, demonstrating a shift in thinking over time [23]. Respondents identified both barriers to and facilitators of providing palliative care, suggesting possible changes, such as training and further education.

Despite this conceptual understanding, the data also showed that this does not necessarily go on to influence direct clinical practice, as some staff were thought to be fearful of introducing palliative care; even with the recent introduction of a specialist palliative care team, some still may not refer. Our study demonstrated that healthcare professionals often lacked acceptance that palliative care input was needed and sometimes were uncertain if no further active treatment options remained. Similar findings were illustrated in a recent study showing delays to palliative care input related to clinical uncertainty, often encountered in children with life-limiting and life-threatening conditions [24]. Stigma associated with palliative care is not a new finding, as previous research [10,17,21,23,25,26,27] involving a series of different healthcare groups, ranging from allied healthcare professionals to doctors, has come to similar conclusions. Barriers preventing referral in these studies included insufficient exposure to palliative care, discomfort discussing the inevitable outlook, negative family attitudes and therapeutic failure. Evidence has shown that palliative care is still perceived by patients, families and healthcare providers as stopping treatment, loss of hope and approaching death [21,28,29,30]. Our study also demonstrated fears surrounding palliative care ranging across societal, staff and family perspectives, arguably stemming from instilled attitudes [31]. An American paper suggested that disparate views from nurses and doctors may hinder efficient collaborative working [32], whilst other studies have identified knowledge gaps among both doctors and nurses [33,34] as a potential barrier. The lack of understanding from staff was also reported in our study, suggesting a possible ingrained attitude towards palliative care.

Our study also explored PICU professionals’ experiences of providing palliative care in the context of working alongside a recently established individual palliative care team. Respondents expressed a need for more robust education to help overcome associated stigmas and misperceptions. In terms of resources, additional funding and staffing were judged to be necessary. It is thought that implementation of education and training may assist in overcoming the misperception of palliative care. More referrals to palliative care teams could reduce fear, symptom burden and hospitalisation, enabling patients to remain safely at home [35,36]. Healthcare professionals agreed that collaborative working and meeting the wishes of the families are thought to be key roles in providing this type of care in conjunction with the palliative care team. In the wider literature, it is thought to be considered a part of daily PICU working, improving both care and satisfaction reported by both patients and their families [37,38,39,40].

To lessen the barriers identified in this study, palliative care teams have the opportunity to act as facilitators by developing working relationships with other teams [41]. To integrate early palliative care, evidence has shown that teams need to be well-staffed and trained and have the resources to accommodate referrals [42]. These requirements present potential challenges, particularly in relation to funding at both an institutional and policy level [23]. For example, in adult palliative care, charities and charitable hospices are the main providers of end-of-life care in the UK, with only around 30% of their income coming from the UK’s governments and NHS sources [43]. Although the UK government has provided emergency funding in the past, this is not a long-term solution [44]. The study findings suggest that improved education, funding, and staff retention could address such barriers. Unfortunately, the issue remains complex and is not easily overcome due to funding source, time, and staff constraints.

### 4.1. Strengths and Limitations

Our study has some limitations. Our response rate was 50% for unknown reasons, failing to capture a large representative sample and meaning that potential perspectives could have been missed. Although smaller than we had hoped, it can still be considered a reasonable response rate when compared to similar qualitative studies [13,45]. It should be noted that, although rich data were captured, they were only gathered from a single site. Therefore, the views and findings may not be representative on a wider level. Our data collection site was also a paediatric tertiary centre with a relatively newly established paediatric palliative care team, limiting the generalizability of the findings to centres with larger, more established teams and those hospitals without specialist palliative care teams. The data lacked participant characteristics as, unfortunately, we did not capture data on age or gender and, additionally, doctors and advanced nurse practitioners were grouped into one survey, making it impossible to delineate specific responses from the two groups. The questionnaire did not undergo a pilot phase, which may have been beneficial in reducing our limitations.

Despite these limitations, the study remains relevant and demonstrates the potential benefits of better integration of PPM into the PICU, adding to the limited UK-based literature. Our study also has a unique viewpoint as data were from a tertiary centre with a palliative care team that is still small and continuing to develop its services across Scotland. Results may help to support other growing palliative care teams to work towards better integration within PICUs through the gaps identified.

### 4.2. Future Research

Future research should further explore PICU professionals’ hesitation when referring to palliative care, as found in this study, but on a larger scale, exploring the views of other specialists who may typically also have a large number of palliative care patients. From the open-ended questions, it was evident that palliative care was valued as part of the PICU. However, agreement on how this could best be delivered was not addressed in this study and requires further research. There is also a need to evaluate existing service models to later determine the most suitable model in line with each hospital environment whilst also factoring in cost and sustainability. Despite five open-ended questions being asked, our results only reflected four as limited evidence was found in answering question three. Future research could explore this question to determine which groups should be referred to a specialist PICU palliative care service.

### 4.3. Implications for Services

This study highlights the urgent need for service development to improve access to specialist palliative care teams, including a re-consideration of current funding streams as these impact future education and training. Currently, specialist palliative care teams are limited by inconsistent services and a lack of training opportunities, and policymakers and service managers need to take action. Throughout this study, respondents advocated for education surrounding palliative care. This could be achieved through changes to undergraduate training and more advanced content in postgraduate training to support healthcare professionals’ knowledge and clinical practice. All of this could then allow for better integration, more timely referral and improved communication with children and their families.

## 5. Conclusions

This study demonstrates that the attitudes, knowledge, meaning and understanding of palliative care are complex and nuanced. Despite findings demonstrating a good understanding of the concepts of palliative care and a willingness to learn, there are on-going barriers to integration of palliative care into the PICU. These still relate to stigma, resources, education and training. If more directed education and training programmes were established, the identified barriers could be addressed. However, this would require extensive funding to encourage staff retention and buy-in from policymakers.

## Figures and Tables

**Table 1 healthcare-11-02438-t001:** Open-ended questions.

Open-Ended Questions	Type of Question
What is your understanding of palliative care?	Open-ended
2. What is/should be the role of palliative care in PICU?	Open-ended
3. What groups do you think should be referred?	Open-ended
4. What are the barriers to the delivery of palliative care in PICU?	Open-ended
5. What is the ideal service model for palliative care within PICU?	Open-ended

**Table 2 healthcare-11-02438-t002:** Participant characteristics.

**Number of Participants Surveyed**	32
**Number of responses**	16
**Occupation**	
	19 consultants
	4 advanced nurse practitioners (ANPs)
	9 nurses

**Table 3 healthcare-11-02438-t003:** Categories, subcategories and codes with frequencies.

Question (Q) the Category Related To	Category Name	Subcategory	Code	Frequency per Code	Frequency per Subcategory	Frequency per Category
Q1. What is your understanding of palliative care?	Role of palliative care	1. Collaborative working	Interdisciplinary team working	12	37	37
Q2. What is/should be the role of palliative care in PICU?			Multidisciplinary team	21		
Q5. What is the ideal service model for palliative carewithin PICU?			Parallel planning	4		
Q1. What is your understanding of palliative care?		2. Family-centred care	Fulfilling wishes	9	28	28
Q2. What is/should be the role of palliative care in PICU?			Holistic care	12		
Q5. What is the ideal service model for palliative care within PICU?			Individualised care	7		
Q5. What is the ideal service model for palliative care within PICU?	Experiences of providing palliative care	1. Education and collaboration	Collaborative working is required	7	35	35
			Knowledge is required	12		
Q4. What are the barriers to the delivery of palliative care in PICU?			Education and training are both barriers and facilitators	16		
Q4. What are the barriers to the delivery of palliative care in PICU?		2. Stigma associated with palliative care	Family fear	3	18	18
			Misperception of what palliative care is	6		
			Societal fear	2		
			Staff fear	7		
Q4. What are the barriers to the delivery of palliative care in PICU?		3. Resources impacting palliative care	Staffing and resources	6	10	10
			Time limitations	4		

## Data Availability

The data presented in this study are available on request from the corresponding author. The data are not publicly available due to privacy.

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
