# Peer review of "Healthcare Professionals’ Attitudes towards and Knowledge and Understanding of Paediatric Palliative Medicine (PPM) and Its Meaning within the Paediatric Intensive Care Unit (PICU): A Summative Content Analysis in a Tertiary Children’s Hospital in Scotland—“An In Vitro Study”"

_healthcare, 2023, doi:10.3390/healthcare11172438_

Round 1
Reviewer 1 Report
Suggestions for revision:
From the methodological point of view, the qualitative study aimed to “explore the experiences of those who provide paediatric palliative care and identify any barriers or facilitators.”. However, in the abstract, the authors refer that the objective was to identify the attitudes, meaning, knowledge and understanding of PPM in the PICU environment (...) and the barriers and facilitators for integrated work.” Thus, it would be important to have consistency between these aspects to clarify the data analysis that was carried out. It is also unclear how the authors developed the Delpy method (expert rounds). They mention that they used five free-text questions, but the categories obtained from the content analysis don’t clearly show results that respond to these questions. They don’t present the results of the five closed questions that the authors refer, in the abstract, they included in the data collection instrument.
Reviewer 2 Report
The article "The Attitudes, Knowledge, Meaning and Understanding of Paediatric Palliative Medicine (PPM) within the Paediatric Intensive Care Unit (PICU): A Summative Content Analysis in a Tertiary Children’s Hospital in Scotland" is a well-studied single-centered study with significant findings. There are many studies in the current topic around the world. This study will serve as another addition highlighting the need for pediatric palliative care (PC) within the pediatric intensive care unit (PICU) with specific consideration of quality issues. The study is important as reference for the policy makers. Although this is a well-studied and well-written manuscript few minor suggestions for improving the manuscript are as follows:
- Please edit the citation from the abstract (line 24).
- The result section in the abstract need to be revised. The first starting sentence is not complete.
Strength
- Validates the need of pediatric palliative care (PC) within the pediatric intensive care unit (PICU). Although the need was known, the study shows gaps in fruitful proper implementation in clinical care. The study highlights that there is still stigma, lack of trainings/knowledge, hindrances (integration, communication, referral etc) during the patient care.
- The results can be used by the policy makers.
- Although the results are from a small sample group, it can be helpful as base line for scaling the study to other sites with meaningful higher participation
- The authors have duly accepted the limitations and need of further study for insightful qualitative conclusion.
Weakness
- The study is single centered with 50% responses from a handful of participants.
- The results are not quantitative more qualitative in nature
- The results are just prospective/understandings of the limited individuals which lacks proper controls.
- The authors themselves agree that the results of this study may not be representative for other sites. (Line 293, 294: “the views and findings may not be representative on a wider level”)
- Authors themselves agree that the study site used in the study has a new PC care establishment which may have influenced the results. (Line 295, 296: “Our data collection site was also a paediatric tertiary centre with a relatively newly established paediatric palliative care team, limiting the generalisability of the findings to those with larger, more established teams and those hospitals without specialist palliative care teams”)
Reviewer 3 Report
The manuscript entitled “The Attitudes, Knowledge, Meaning and Understanding of Paediatric Palliative Medicine (PPM) within the Paediatric Intensive Care Unit (PICU): A Summative Content Analysis in a Tertiary Children’s Hospital in Scotland” discusses a relevant topic. However, the authors should consider some important points to improve the manuscript.
1. Title
I suggest to change to “Healthcare Professionals Attitudes, Knowledge, Meaning and Understanding of Paediatric Palliative Medicine (PPM) within the Paediatric Intensive Care Unit (PICU): A Summative Content Analysis in a Tertiary Children’s Hospital in Scotland “an in vitro study”
2. Abstract:
The abstract is well-structured and adequate. I suggest to remove the reference to the management software, NVivo. It would be better to replace the text by “from the data of a management software”.
3. Introduction
This section is well-written. The references are current and provide a good basis for the central question of the study.
4. Material and Methods
Please, remove the text “It was hoped that the results would assist in future implementation of services and provide us with a better understanding of providing this care.” This statement is more appropriate to Discussion section.
The authors didn’t mention if there was any specialist palliative professional in the sample. It is relevant to include some information about it in order to avoid bias.
I believe that ethical approval is necessary because the study describes healthcare professionals opinions. They are the subjects of the research.
The authors mentioned “5 open-ended and 5 closed questions” in the abstract, but the 5 closed questions didn’t appear in the Methods.
5. Results
Excellent
6. Discussion
The Discussion is well-structured and the authors mentioned the limitations of the study as well as implications for services.
Round 2
Reviewer 1 Report
The revised version of the manuscript is sufficiently improved for publication.
Reviewer 3 Report
The authors added all the sugestions to the final version of the manuscript. We think that it is suitable for publication now.